# Secondary Oral Cancer after Systemic Treatment of Hematological Malignancies and Oral GVHD: A Systematic Review

**DOI:** 10.3390/cancers14092175

**Published:** 2022-04-27

**Authors:** Aleksandra Janowiak-Majeranowska, Jakub Osowski, Bogusław Mikaszewski, Alan Majeranowski

**Affiliations:** 1Department of Otolaryngology, Faculty of Medicine, Medical University of Gdańsk, 80-210 Gdańsk, Poland; bemi999@wp.pl; 2Students’ Scientific Association, Faculty of Medicine, Medical University of Gdańsk, 80-210 Gdańsk, Poland; jakub.osowski@gumed.edu.pl; 3Intercollegiate Faculty of Biotechnology, The University of Gdańsk and the Medical University of Gdańsk, 80-210 Gdańsk, Poland; a.majeranowski@gumed.edu.pl

**Keywords:** oral cancer, hematological treatment, hematopoietic cell transplant, graft-versus-host disease

## Abstract

**Simple Summary:**

The number of hematopoietic cells transplantations is increasing annually, and the average survival time after treatment is gradually extending. As a result, these patients experience late side effects of hematological treatment, including secondary oral cancer. The oral cavity is relatively easily accessible for examination; therefore, every physician should be familiar with different pathologies of this area and, when necessary, refer patients to an ENT specialist for examination. In this review, the authors tried to identify a potential correlation between the oral chronic form of graft-versus-host disease and oral cancer in patients after hematopoietic cell transplantation, and they tried to propose a surveillance protocol and tips that can be used during a patient’s follow-up.

**Abstract:**

(1) Purpose: In this article, the authors decided to systematically review the available literature to identify potential correlations regarding secondary oral carcinoma occurring after hematological systemic treatment and oral chronic graft-versus-host disease. (2) Methods: Medline (PubMed) and Scopus (Elsevier) databases were searched, including articles from the years 2002–2022. The 33 unique results were assessed by a PRISMA flowchart, and we rejected 24 papers and included 9 articles in the review. (3) Results: The majority of patients suffered from the oral form of chronic graft-versus-host disease before the diagnosis of oral malignancy. Two common cancer sites were the tongue and buccal mucosa. The exact percentage of secondary oral carcinoma after hematopoietic stem cell transplantation could not be estimated due to a lack of data. (4) Conclusions: Every physician taking part in the follow-up of patients after hematological treatment should be aware of the possibility of secondary neoplastic disease in the oral cavity, especially in patients with oral graft-versus-host disease. Proper follow-up protocols and monitoring are needed in this patient group as the cause of these cancers appears to be multifactorial.

## 1. Introduction

Secondary cancer is defined as an independently developing cancer in a person previously diagnosed with a different type of malignant tumor. Secondary cancer is not a metastatic tumor, but it is a neoplasm of a completely different histopathological type, which can occur at different times from the time of primary diagnosis. If the secondary tumor occurs within less than 6 months, it is referred to as a synchronous tumor, and if it occurs after 6 months, it is referred to as a metachronous tumor [1].

In recent years, the frequency of hematopoietic cell transplantation (HCT) has increased annually [2]. HCT is used in the systemic treatment of malignant (such as leukaemia or lymphoma) as well as nonmalignant (such as thalassemia or aplastic anemia) hematological conditions [3]. However, like any other therapeutic intervention, it has some specific complications.

The graft-versus-host disease (GVHD) is one of the most common complications of HCT. The pathogenesis of this phenomenon involves the attack of donor T lymphocytic cells on the recipient’s tissues, each of which have a different major histocompatibility complex (MHC). GVHD is divided into two major types: acute and chronic. The classic criteria to classify GVHD is based on the time of disease onset, with the acute form beginning during the first 100 days after transplantation and the chronic form beginning after 100 days. Currently, however, the diagnosis of these forms is based on clinical symptoms [4]. Acute GVHD usually manifests as a triad of symptoms comprising liver dysfunction (often presenting as jaundice), various skin lesions, and gastrointestinal symptoms, such as abdominal pain and diarrhea [5]. On the other hand, chronic GVHD (cGVHD) may affect nearly every organ system, with the following most commonly involved: the skin, eyes, mouth, gastrointestinal tract, liver, lungs, and musculoskeletal and genitourinary systems. It is pathophysiologically and clinically similar to autoimmune diseases, leading to a gradual deterioration of the patient’s quality of life and increased mortality risk [4].

This systematic review aimed to identify potential correlations between the oral form of cGVHD and secondary oral carcinoma (SOC) after HCT. SOC is defined as an oral cancer that develops in patients (who previously did not have this type of cancer) after an HCT procedure.

## 2. Materials and Methods

### 2.1. Search Strategy

In order to establish the association of chronic oral graft-versus-host disease with secondary oral cancer, a systematic review was performed in accordance with the Preferred Reporting Items for Systematic Reviews and Meta-Analyses (PRISMA) method [6]. Electronic databases of Medline (PubMed) and Scopus (Elsevier) were used to identify articles pertinent to the purposes of the review. Articles published in the period between 2002 and 2022 were sought from the databases. The review protocol was registered on Research Registry (review registry 1295).

### 2.2. Searching and Data Screening

On 25 February 2022, the abovementioned databases were manually searched by two independent authors (A.J.M., J.O.). The titles, abstracts, and keywords were searched for common terms used to described oral cancer (oral carcinoma) in combination with various terms for bone marrow transplant (hematopoietic cell transplant) or in combination with the term “graft versus host disease”. The search strategy used is presented in Appendix A. The initial search identified a total of 33 articles, of which 24 were excluded due to various reasons and 9 were included in the review. The entire screening process is detailed in a PRISMA flowchart in Figure 1. Any disagreements between the two independent authors were resolved through deliberation and articles were included following a consensus.

### 2.3. Eligibility Criteria

The systematic review was based on a PICOTS (population, intervention, comparison, outcome, time, and setting) format question [7]. Population: patients with hematological diseases who underwent systemic treatment and hematopoietic stem cell transplantation or bone marrow transplantation. Intervention/exposure: patients with oral form of cGVHD. Comparison: patients without oral cGVHD; outcome was the diagnosis of a secondary oral cancer (SOC). As SOC is a late complication of HCT, surveillance time should be 10 years and more after procedure. The setting was all physicians, especially ENT specialists involved in follow-up of patients after hematopoietic stem cell transplant. The inclusion and exclusion criteria are presented in Table 1.

### 2.4. Data Extraction and Processing

From each included article, the following data were manually extracted: the total number of the HCT cohort, the number of people who developed GVHD after HCT, the number of people who developed secondary oral cancer, the number of people with GVHD before the onset of cancer, the most common locations of SOC (also presented as a percentage), and the intensity of the GVHD process. If relevant data were available, the percentage of people undergoing HCT who developed GVHD and the percentage of people who had SOC coexisting with oral cGVHD were calculated. Data on secondary neoplasms of parotid gland and maxillary bone in Kruse et al. as well as on supraglottic carcinomas of the larynx and oropharyngeal carcinomas in Douglas et al. were not considered for the purposes of this review [8,9]. As these locations are not classified as the oral cavity, including them in the overall result would make the calculations liable to either over- or underestimation of results and false conclusions.

The quality of the included studies was assessed using the Newcastle–Ottawa Scale [10]. This scale was created to evaluate nonrandomized, observational studies (cohort and case–control studies). A maximum of nine stars can be awarded to a study for each of the following items: selection (four stars), comparability (two stars), and outcomes (three stars). A study is considered to be of high quality if it scores seven or more stars. Kruse et al. was assessed with AMSTAR-2 criteria [11], as these criteria are created for evaluation of review articles. AMSTAR-2 is not designed to generate a point score; it is intended to generate the overall rating by assessing critical domains of the review.

## 3. Results

### 3.1. Databases Search Results

Thirty-three unique results were obtained through manually search of aforementioned databases. Following title and abstract screening, 15 articles were excluded as they did not meet the PICOTS criteria, 18 articles were considered for full-text assessment. Of these, two papers were not retrieved, and therefore, they were excluded from the review. From the remaining 16 papers, a total of 9 articles were excluded; >5 were case reports, 1 was a letter to the editor, 1 was a case series, and 2 described patients suffering from Fanconi anemia (an independent risk factor for the development of oral cancer). Finally, seven articles were considered eligible for inclusion in the review.

### 3.2. Included Studies Characteristics

Six retrospective cohort studies and one review article were included in the review.

Atsuta et al. described a cohort of 17,545 adult (defined as older than 16 years) HCT recipients in the Japanese population, from which the 269 who developed secondary solid organ cancer (including 64 oral cancers) were selected. In a retrospective analysis by Douglas et al., patients from one center (Toronto) who developed GVHD or secondary oral cancer after HCT were described. Patients suffering from Fanconi anemia or immunodeficiency were excluded due to known increases in carcinogenesis risk. A total of 23 secondary oral malignancies were described, with no total number of patients treated with HCT. The purpose of a retrospective study by Hanna et al. was the analysis of genetic alterations in secondary oral carcinomas occurring after HCT. It included a cohort of 31 people who received hematology treatment at Boston. Despite the different assumptions of this study, it contained enough patient characterization data to be included in this review. A group of 80 patients with oral complications (acute and chronic GVHD, dysplastic lesions, and SOC) after HCT was collected by Leuci et al. in a retrospective cohort analysis conducted at the University Hospital of Naples. This cohort included seven patients with a secondary tumor. A retrospective analysis led by Mawardi et al. included data from three centers (Boston, Jerusalem, and Campinas, Brazil). The paper mainly focused on the description of changes occurring in the oral cavity after transplantation; out of 26 patients included, 18 neoplasms were described. The exact number of patients with GVHD and SOC was not specified; the number of GVHD cases was related to the total number of patients with dysplasia and cancer. The most detailed retrospective study by Santarone et al. thoroughly described a group of 908 people receiving allo-HCT at a center in Pescara, Italy. Patients with Fanconi anemia and immunodeficiency were not included. SOC was diagnosed in 12 patients. The only review article was that of Kruse et al. Over the span of thirty years (1978–2008), a group of 64 patients with secondary oral cancer was identified (from other articles like case reports and retrospective studies). A table containing clinical data of each patient was attached to the article and available for download from a journal site. Data from this compilation were used for this systematic review after the removal of neoplasms in locations other than the oral cavity in the head and neck.

In total, 208 patients with SOC were included in the systematic review.

### 3.3. Risk of Bias Assessment

Each article included in the review was separately assessed using the Newcastle–Ottawa Scale; only the paper by Kruse et al. was assessed with AMSTAR-2 criteria (Table 2). All of the included retrospective articles were of high quality. The most crucial source of bias, in the majority of the studies, was due to the inadequate and short follow-up time (median time in months: Hanna et al. 22.4 [12]; Leuci et al. 22 [13]; Mawardi et al. 21.5 [14]). Atsuta et al. and Douglas et al. had slightly longer follow-up times (minimum 5 years and minimum 2 years, respectively) [9,15]. Secondary cancers are late complications of HCT and have a tendency to reoccur; therefore, it is essential to have at least 10 years or more of follow-up. Only Santarone et al. describe an adequate long-term follow-up (median follow-up of 17 years). The majority of the articles clearly described the presence or absence of oral cGVHD. However, Shah et al. and Mawardi et al. lacked information related to cGVHD. These reports included the total number of GVHD cases—for dysplasia and for cancer—without differentiating between groups. Similarly, information regarding SOC site was lacking for a majority of the included patients. The main source of bias in Kruse et al. was the fact that it was not a systematic review (only one database was searched by one author, but the process of study selection was described, and a risk of bias assessment was not conducted). The quality of this review article was moderate.

### 3.4. Results of Data Synthesis

The data extracted from the included articles are presented in Table 1. A total of 208 patients who developed SOC were described in the included literature. One hundred and ninety (91.35%) of them had a definite presence or absence of oral cGVHD. Most of them, 136 (65.38%), had oral chronic GVHD before cancer, whereas in 54 (25.96%) people, cancer developed without prior cGVHD. The remaining 18 patients (8.65%) did not have any exact information about the coexistence or absence of GVHD (Table 3). For 94 (43.52%) of the 221 patients, information on the exact localization of SOC was not described. The tongue (61 patients, 28.24%) and the buccal mucosa (23 patients, 10.64%) were found to be the most common sites for SOC in this cohort. Other less frequent sites included the lip (12 patients, 5.56%), alveolar process (9 patients, 4.17%), gingiva (6 patients, 2.78%), hard palate (5 patients, 2.31%), retromolar trigone (4 patients, 1.85%), and floor of the mouth (2 patients, 0.93%). The overall results are presented in Table 4. The percentages did not add up to 100 because in articles by Mawardi et al. and Kruse et al., cases of multiple tumor lesions in the oral cavity of the same patient were described. 

## 4. Discussion

The number of HCTs has been increasing annually along with an improving average survival time post-HCT [2]. As shown in this review, the risk of developing SOC also increases with time post-transplantation [17,18]. This is especially pertinent for pediatric patients undergoing HCT, who are likely to have long-term survival [19]. Therefore, the occurrence of SOC may show an increasing tendency. It must be noted that the symptomatology of neoplastic lesions in the oral cavity is very similar to the chronic form of oral GVHD, which occurs much more frequently (almost 50% of patients after HCT develop chronic GVHD within a year) [20] than secondary cancers of the oral cavity. Both conditions may present as lichenoid changes, atrophy of mucosa, ulcerations, and dysplastic changes. In addition, the oral form of cGVHD is a complication that reduces the patient’s quality of life due to pain, difficulty eating, and decreased salivation [21]. This review identified that the majority of patients with previously diagnosed oral cGVHD later developed SOC. In the literature, the oral form of cGVHD is recognized as a risk factor for oral cancer [8,14,15,18,22,23,24,25]. The diagnosis of this autoimmune reaction may reduce a physician’s vigilance [23]. The persistence of lesions despite proper treatment and the appearance of new lesions in the oral cavity should be considered indications for another biopsy.

In 2014, a new version of the oral cGVHD severity classification developed by the National Institutes of Health was published [26]. Most of the articles presented in this review provided information regarding the presence of cGVHD. However, only a few studies described the severity of GVHD [12,15,16]. In order to fully understand the potential relationship between the severity and appearance of lesions and secondary cancer of the oral cavity, it is advisable to describe the changes found according to a standardized scale, in addition to noting the exact stage and providing a detailed description of the changes. Kruse and Gratz also developed a new classification of findings in the oral cavity occurring after HCT in order to help identify high-risk patients [8]. The classification consists of Grades 0–3: (0) no involvement, (1) erythema an/or hyposalivation, (2) lichenoid appearance, (3) ulceration/tumor. However, its use in clinical practice needs to be assessed in a randomized trial.

In the treatment of cGVHD, immunosuppressants are used, which are themselves proven to promote carcinogenesis. For instance, the development of secondary neoplasms after GVHD treatment with cyclosporine was reported [25]. Moreover, there are studies showing the influence of immunosuppression (after kidney transplantation) [27] on the increased risk of HPV infection, of which oncogenic types (especially 16 and 18) are an established risk factor for squamous cell carcinoma [28]. Therefore, the effect of immunosuppression on the increased risk of SOC appears to be multifactorial.

In SOC occurring as a complication of GVHD, Tamma et al. described that there is an enhanced inflammatory response (infiltration of T and B lymphocytes, mast cells, and macrophages) and an increase in vasculogenesis (expressed as an increase in CD31) [29]. On the other hand, Hayashida et al. described an increase in Th1 lymphocytes with IL2 and IFNy, as well as Th2 lymphocytes with IL4 and IL5, in tissues affected by the oral GVHD process [30]. These changes, as well as the alterations described in other articles (such as those previously mentioned by Hanna et al. [12]), may form the basis for implementing targeted therapy in the future.

Ionizing radiation damages DNA, causing cell mutations, which may contribute to the initiation of the process of carcinogenesis. There is a dose-dependent increase in the number of damaged cells. Radiotherapy of the head and neck area can cause a secondary neoplasm in this anatomical site. This has been well-documented in the literature describing the development of SOC after radiotherapy for nasopharyngeal cancer. Moreover, it has been shown that the type of radiotherapy influences the likelihood of developing SOC [31,32,33,34,35,36]. Such secondary tumors have also been defined by the term RISCCO (radiation-induced second primary squamous cell carcinoma of the oral cavity) [36]. Additionally, Hashibe et al. found that radiotherapy of head and neck tumors not only increases the risk for secondary neoplasms in this region but also increases the risk in several other organs [37]. However, there is a lack of clear guidelines regarding the use of total body irradiation (TBI) during conditioning due to published reports providing conflicting evidence regarding its value as a risk factor for SOC [15,22,36,37,38]. It should be noted that the use of TBI and allogenic transplantation of hematopoietic cells from an unrelated donor are also well-known risk factors for GVHD [4,13], indicating that the influence of these factors on the formation of SOC may be synergistic.

Of the included studies, Mawardi et al. had a relatively high percentage of patients using alcohol and tobacco [14], while Chaulagin et al. and Leuci et al. described SOC cases in HCT patients without histories of alcohol or tobacco use [13,22]. Alcohol, tobacco, betel chewing, low socioeconomic status, and infection with oncogenic HPV types are classic risk factors for oral cancer occurrence [28]. It is plausible that these factors further impact SOC development in post-HCT patients synergistically with the other factors discussed above. However, there is still a lack of precise data, with included studies presenting limited explanations regarding these factors. Therefore, further research on the influence of these factors on SOC post-HCT is essential.

Another risk factor for oral cancer is lichen planus. It is an autoimmune disease mediated by T lymphocytes against basal epithelial keratinocytes [39]. Morphologically, it may resemble the changes occurring in the oral form of chronic GVHD [21]. However, there are disputes in the literature regarding its potential for malignant transformation. Halonen et al. conducted a retrospective study, in the Finnish population, highlighting that the presence of lichen planus correlates with an increased risk of cancer in the mouth, lip, tongue, esophagus, and larynx (without increasing the risk of throat or skin cancers). However, the majority of cancer diagnoses in this cohort occurred during the first year of follow-up, suggesting that seeking medical attention was associated with cancer symptoms, not with the presence of lichen planus [40]. In the Japanese population, Tsushima et al. found a slight increase in the risk of malignancy (0.7%) people with oral lichen planus [41]. Since the oral cavity is a readily accessible examination site, it should be regularly examined in patients with lichen planus.

In 2020, the new classification of oral potentially malignant disorders (OPMDs) by World Health Organization Workgroup was published. OPMDs are oral lesions of varying morphology and appearance and are mono- or multifocal, and they may progress to cancer or cause regressions. Classically, leukoplakia (white patches), erythroplakia (red patch), and oral lichen planus (OLP) were considered to be OPMDs. In the latest classification, two types of changes were added—the oral form of chronic GVHD and oral lichenoid lesions (OLLs), defined as changes resembling OLP but not fulfilling its clinical and histopathological criteria. These lichenoid lesions may be caused by drugs or dental restoration. Lesions occurring in OLP, OLL, and oral GVHD may be very similar; therefore, they require thorough diagnostic processes, including biopsies and histopathological examinations [42].

Locoregional recurrence tends to be a major challenge in patients with SOC. Early stage (T1–T2, N0) oropharyngeal carcinoma can be treated with primary surgery with (chemo)radiotherapy or radical (chemo)radiotherapy [43]. In a phase I, single-center, prospective clinical study, it was found that selective neck dissection (SND) after trans oral robotic surgery (TORS) of oral or oropharyngeal carcinoma provided using adjuvant chemo- or selective radiotherapy of a lesser intensity maintained a reduction in locoregional recurrence rates [44]. Similar conclusions were made by Meccariello et al., who noted that SND after TORS is an effective method [45].

The basis of a surveillance protocol should be cooperation with the patient. The patient should be informed regarding the nature of the disease, treatment options, and the predictable complications. Such knowledge allows them to react and seek appropriate help in the event of possible complications. An interdisciplinary approach is crucial for patient education as it enables physicians to provide accurate information and clarify doubts, using various information materials such as leaflets and websites, and end with support groups and contact with patients with similar diseases. Unfortunately, some patients may not be able to grasp medical information sufficiently, which may adversely affect the implementation of such recommendations. Therefore, it is also crucial to provide complex information in an easily understandable manner for the patient and their relatives.

Self-examination of the oral cavity may appear to be the first element of patient surveillance. However, research shows that despite individual instruction, it is ineffective in detecting potentially malignant changes [46]. Such a relationship is not observed in the case of follow-up examinations by health care professionals. Oral examination via a specialist is able to detect pathological changes with a greater frequency, translating into reduced mortality [47]. As technology develops, it will be possible to use dedicated smartphone applications, enabling patients to screen a self-detected change, thereby improving the screening process [48]. The guidelines for screening described by Santarone et al. should be used [16].

It is also pertinent that surveillance occurs within a single center that follows the patient’s progress after transplantation. A lack of cooperation between different centers may delay the diagnosis of neoplastic disease. Cancers occurring after HCT are more aggressive and are associated with a worse prognosis and higher recurrence rates than primary oral cancers, making appropriate patient follow-up pertinent [14,22]. Smaller hospitals may not have enough experience, as they have contact with similar patients much less frequently than large oncology centers. In the modern world, technology can be utilized for archiving photographic documentation from examinations, allowing more precise monitoring of the appearance and progression of changes. Wherever possible, photographic documentation should be used. However, financial limitations in low- and middle-income countries may prevent the adoption of such recommendations. Additionally, several programs, neural networks, and computer algorithms have been developed for the purpose of detecting squamous cell carcinoma through photographic evidence, with a sensitivity of 91% and a specificity of 93.5% [49]. Another group developed a neural network for detecting oral cancer in photos taken with a regular smartphone, with a sensitivity of 83.3% and a specificity of 96.6% [48]. However, a systematic review and meta-analysis conducted by Mazur et al. (34 articles were included) proved that in vivo imaging-based techniques (narrow band imaging, vital staining colorants, optical spectroscopy, autofluorescence, and endoscopy) cannot replace biopsies in the early detection of oral diseases [50]. Tissue biopsy and histopathological examination remains a “gold-standard” for the early diagnosis of OPMDs and oral carcinoma.

## 5. Conclusions

In the majority of patients with SOC (65.38%), tumor development was preceded by the oral form of cGVHD. A precise description of GVHD status was available for most of the patients (91.35%). However, due to limited descriptions of GVHD severity in the included literature, this review was unable to determine the exact lesions in the oral cavity with the highest risk of malignant transformation. The common places for SOC occurrence were the tongue (28.24%) and the buccal mucosa (10.64%), although exact location information for a significant proportion of patients (43.52%) was unavailable; therefore, the estimation of the location was burdened with a moderate risk of statistical error. On the basis of the collected data, it was not possible to estimate the exact percentage of people who developed SOC after HCT due to a lack of information in the included articles (an attempt to estimate would be characterized by a high risk of statistical error). There is a need for a large, multicenter, prospective study assessing the discussed parameters to better assess the correlation of this common HCT complication with the development of secondary oral cancer. Such a study should note the total number of patients undergoing HCT (including the percentage of patients who developed GVHD), the number of patients with SOC, the percentage of patients with chronic oral GVHD prior to SOC, detailed descriptions of the severity and morphology of changes in the oral cavity, and the exact location of the SOC. However, since the cause of secondary oral cancer appears to be multifactorial, it is challenging to take each factor into account. Classical, well-documented risk factors and HCT-specific factors, such as TBI during conditioning, cGVHD, and subsequent immunosuppression, could have a synergistic impact on the development of a malignant lesion. Every physician taking part in the follow-up of HCT patients should be aware of the possibility of secondary neoplastic disease in the oral cavity.

## Figures and Tables

**Figure 1 cancers-14-02175-f001:**
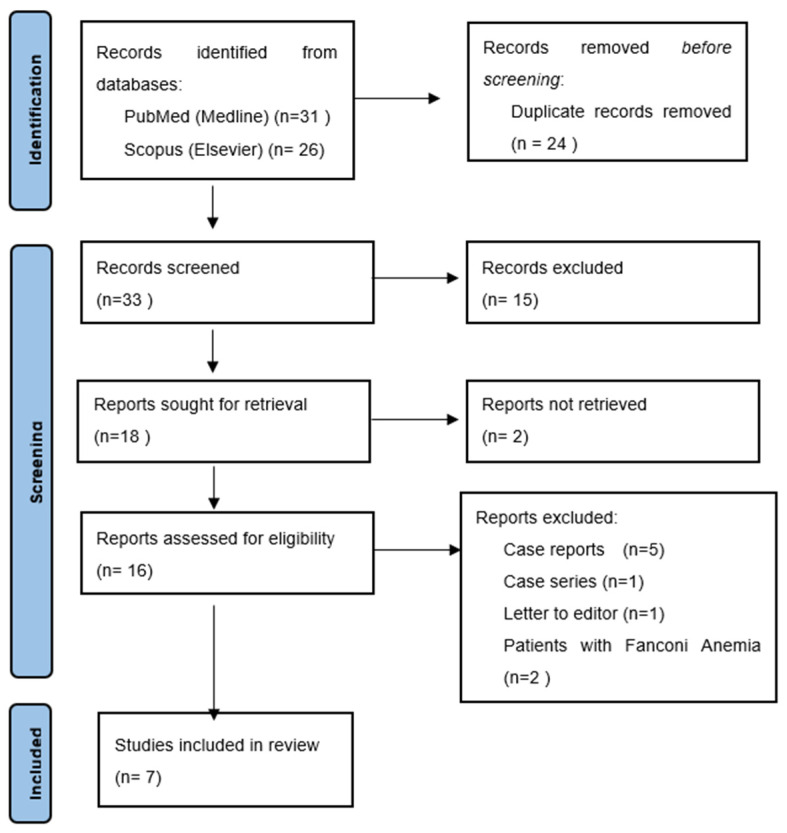
PRISMA flowchart.

**Table 1 cancers-14-02175-t001:** Inclusion and exclusion criteria for selection.

Inclusion Criteria	Exclusion Criteria
History of HCT	Case report/case series article
Diagnosis of SOC	Fanconi anemia

HCT—hematopoietic cell transplant; SOC—secondary oral cancer.

**Table 2 cancers-14-02175-t002:** Data extracted from articles.

Article	Total Patients (*n*)	Patients with cGVHD (*n*)	HCT Patients and GVHD (%)	Patients with SOC (*n*)	Patients with SOC and GVHD (*n* (%))	Most Common SOC Site	GVHD Severity (*n*)	Newcastle–Ottawa Score (*n*)/AMSTAR-2 Criteria ^1^
Atsuta et al. [15]	17,545	NR	NR	64	39 (60.10%)	NR	Limited (10), Extensive (29)	8
Douglas et al. [9]	NR	NR	NR	23	18 (83%)	Tongue (13), buccal (3), alveolus (3), palate (3), lower lip (1)	Not specified	7
Hanna et al. [12]	NR	NR	NA	31	20 (65%)	Tongue (14), buccal (5), retromolar trigone (4), alveolar (3), palate (3), floor of the mouth (2)	NIH 1 (12), NIH 2 (4), NIH 3 (4)	7
Kruse et al. [8]	NR	NR	NR	53	42 (79.20%)	Tongue (16), lip (6), buccal (3) Not specified (30)	NR	Moderate
Leuci et al. [13]	NR	NR	NR	7	7 (100%)	Tongue (2), buccal (2), alveolar ridge (1), gingiva (1), lip (1)	NR	8
Mawardi et al. [14]	NR	NR	NR	18	NR	Tongue (10), buccal (7), gingiva (4), lower lip (3), alveolar (2), palate (1)	NR	7
Santarone et al. [16]	908	767	84,5%	12	9 (75%)	Tongue (6), buccal (3), lip (1), palate (1), gingiva (1)	Extensive (7), Limited (2)	9

GVHD = graft-versus-host disease, HCT = hematopoietic cell transplant, SOC = secondary oral carcinoma, NIH = National Institutes of Health, NR = not reported. ^1^ AMSTAR-2 was used for assessing Kruse et al.’s article.

**Table 3 cancers-14-02175-t003:** Coexistence of oral chronic GVHD before diagnosis of SOC.

Oral cGVHD Coexistence	*N* (%)
Yes	136 (65.38)
No	54 (25.96)
Not specified	18 (8.65)

**Table 4 cancers-14-02175-t004:** Secondary oral cancer localization.

SOC Localization	*N* (%) ^1^
Tongue	61 (28.24)
Buccal mucosa	23 (10.64)
Lip	12 (5.56)
Alveolus	9 (4.17)
Gingiva	6 (2.78)
Hard palate	5 (2.31)
Retromolar trigone	4 (1.85)
Floor of the mouth	2 (0.93)
Not specified	102 (43.52)

^1^ The sum of the percentage is not 100 since Mawardi et al. and Kruse et al. described patients with multiple sites of oral cancer.

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
