# Peer review of "Secondary Oral Cancer after Systemic Treatment of Hematological Malignancies and Oral GVHD: A Systematic Review"

_cancers, 2022, doi:10.3390/cancers14092175_

Round 1

Reviewer 1 Report

Authors answered to all my requests.

Author Response

Reviewer 1 in his comment stated, that authors answered to all his requests.

Reviewer 2 Report

  • line 164, no description of treatment option has been performed. Currently determine the regional recurrence rate of node-positive oropharyngeal squamous cell carcinoma (OPSCC) in patients undergoing transoral robotic surgery (TORS) and selective neck dissection (SND) followed by observation, radiation, or concurrent chemoradiation are the main goals of treatment. SND after TORS resection of primary OPSCC enables the use of selective and deintensified adjuvant therapy to reduce regional recurrence rates. please cite doi:10.1002/lary.21021 and doi:10.1016/j.anl.2021.05.007
  • PICOTS Framework should be applied.
  • Oral potentially malignant disorders (OPMDs) are lesions that may undergo malignant transformation to oral cancer. The early diagnosis and surveillance of OPMDs reduce the morbidity and mortality of patients. Diagnostic techniques based on medical images analysis have been developed to diagnose clinical conditions.   None of the currently other techniques based on assessing oral images can replace the biopsy. doi:10.3390/ijerph182211775

Author Response

Response to Reviewer 2 Comments:

Point 1: "Line 164, no description of treatment option has been performed. Currently determine the regional recurrence rate of node-positive oropharyngeal squamous cell carcinoma (OPSCC) in patients undergoing transoral robotic surgery (TORS) and selective neck dissection (SND) followed by observation, radiation, or concurrent chemoradiation are the main goals of treatment. SND after TORS resection of primary OPSCC enables the use of selective and deintensified adjuvant therapy to reduce regional recurrence rates. please cite doi:10.1002/lary.21021 and doi:10.1016/j.anl.2021.05.007"

Response 1: Paragraph regarding the newest treatment options, including the transoral robotic surgery and selective neck lymph node dissection has been added to the dicussion section. Authors decide, that this section is more suitable  for informations about newest treatment strategies than the "risk of bias assessment" paragraph. The two articles mentioned above has been cited.

Point 2: "PICOTS Framework should be applied".

Response 2: PICOTS Framework was applied to the article in 2.3 section (Eligibility criteria). T (time) should be 10 years and more after procedure, as secondary oral cancer is late complication of HCT, while S (setting) - all physicians, especially ENT specialists involved in follow-up of patients after hematopoietic stem cell transplant. Population, Intervention/Exposure, Comparision and Outcome remained the same as in previously used PICO Framework.

Point 3: " Oral potentially malignant disorders (OPMDs) are lesions that may undergo malignant transformation to oral cancer. The early diagnosis and surveillance of OPMDs reduce the morbidity and mortality of patients. Diagnostic techniques based on medical images analysis have been developed to diagnose clinical conditions.   None of the currently other techniques based on assessing oral images can replace the biopsy. doi:10.3390/ijerph182211775"

Response 3: The article " In Vivo Imaging-Based Techniques for Early Diagnosis of Oral Potentially Malignant Disorders—Systematic Review and Meta-Analysis" has been cited in the end of last paragraph in Discussion, right after the fragments about smartphone-based imaging assesment algorithms. Conlusion that the tissue biopsy and histopathological examination is the "gold-standard" in diagnosing of OPMDs and oral carcinoma was clearly stated.

This manuscript is a resubmission of an earlier submission. The following is a list of the peer review reports and author responses from that submission.

Round 1

Reviewer 1 Report

It is an original and interesting work.

Even if there is a considerable amount of data, they are still usable. 

The survey methodology is clearly described.

It would be advisable to indicate the total number of patients considered in the study and the related epidemiological data, perhaps with summary tables.

Reviewer 2 Report

Majeranoska et al, in their manuscript entitled “ Secondary oral cancer after systematic treatment of haematological malignancy : a systematic review” , conducts a systematic review of the literature about the potential risks in the secondary oral cancer after hematological systemic treatment, specially with GVHD:

Authors did not describe accurately the search methodology

The manuscript cannot be accepted before major revisions listed below:

  1. How did they create the database ?
  2. How many authors selected the data?
  3. Please include a flow chart of the study selection process.
  4. Include a Table with all inclusion and exclusion criteria of selection.
  5. Include a section in methods with Eligibility criteria

Reviewer 3 Report

Although the topic investigated is relevant, this systematic review has not been designed or conducted according to high standards. 

The study design presents a high risk of bias with low internal validity, receiving  a critically low quality (following the scoring system validated by the influential AMSTAR2 tool). PRISMA reporting guidelines have not been followed (only the flow diagram). A study protocol on the methodological design of this systematic review has not been registered a priori (mandatory to report code and date, prior to publication). This is a fundamental step that is currently considered as an important source of bias, to increase the transparency and integrity of the methods, and decrease the risk of bias of a systematic review. Furhtermore, a qualitative assessment analysis across primary level studies included has not been carried out, which also gives serious limitations to consider this paper as a true systematic review (an essential step to assess the quality of the evidence according to the GRADE initiative). Honestly this study could also be considered as a narrative review in which a flow chart was simply generated and a material and methods section was incorporated into the manuscript (thus attempting to be classified as a systematic review). If the only methodologically acceptable point of this work could be the study selection process, it should be noted that only one database was consulted, when according to Cochrane Collaboration criteria at least 2 databases should be used, Embase being an imperative option. Nothing about the data management and extraction process, inter-evaluator interaction system, clinical, methodological or statistical heterogeneity, publication bias, etc. All these issues should have been imperatively addressed in a systematic review that intends to be published in a prestigious journal, such as our journal Cancers.

In relation to the key topic, the objectives were not clearly defined lead to confusion when the authors started to address the different oral potentially malignant disorders (with references that were not updated, a new WHO workshop on oral cancer and oral potentially malignant disorders having been held last year). Finally, the relationship between chronic graft-versus-host disease and oral lichen planus was also neglected, being this clinical context of paramount importance in these patients, who on several occasions mimic a picture of oral lichen planus, with also a considerable rate of malignant transformation to oral cancer. It should also has been considered as secondary to the therapeutic process of hematologic malignancy.